# Research on a Phase-Shift-Based Discontinuous PWM Method for 24V Onboard Thermally Limited Micro Voltage Source Inverters

**DOI:** 10.3390/mi16101128

**Published:** 2025-09-30

**Authors:** Shuo Wang, Chenyang Xia

**Affiliations:** School of Electrical Engineering, China University of Mining and Technology, University Road No.1, Xuzhou 221116, China; chyxia@cumt.edu.cn

**Keywords:** discontinuous PWM (DPWM) method, phase shift angle, micro three-phase voltage source inverter (3ph-VSI), wide power factor range, switching losses

## Abstract

This research explores a phase-shift-based discontinuous PWM method used for 24 V battery-powered onboard micro inverters, which are critical for thermally limited applications like micromachines, where efficient heat dissipation and compact size are paramount. Discontinuous pulse width modulation (DPWM) reduces switching losses by clamping the phase voltage to the DC bus in order to improve inverter efficiency. Due to the change in power factor at different operating points from motors or the inductor load, the use of only one DPWM method cannot achieve the optimal efficiency of a three-phase voltage source inverter (3ph-VSI). This paper proposes a generalized DPWM method with a continuously adjustable phase shift angle, which extends the six traditional DPWM methods to any type. According to different power factors, the proposed DPWM method is divided into five power factor angle intervals, namely [−90°, −60°], [−60°, −30°], [−30°, 30°], [30°, 60°], and [60°, 90°], and automatically adjusts the phase shift angle to the optimal-efficiency DPWM mode. The power factor is calculated by means of the Synchronous Reference Frame Phase-Locked Loop (SRF-PLL) method. The switching losses and harmonic characteristics of the proposed DPWM are analyzed, and finally, a 24 V onboard 3ph-VSI experimental platform is built. The experimental results show that the efficiency of DPWM methods can be improved by 3–6% and the switching loss can be reduced by 40–50% under different power factors. At the same time, the dynamic performance of the proposed algorithm with a transition state is verified. This method is particularly suitable for miniaturized inverters where efficiency and thermal management are critical.

## 1. Introduction

With the trend toward miniaturization and integration in micromachine drive systems, the demand for high-efficiency, low-loss power converters has become increasingly important [1,2,3]. The proliferation of compact motor drives in applications such as automotive auxiliaries, drones, microfluidic micromachines [4,5], and portable robotics has created a critical need for small-form-factor, onboard power conversion. A 24V battery-powered inverter is a common standard for driving these low-voltage motors directly from a vehicle’s electrical system [2]. The primary challenge for such an inverter is managing significant switching losses and thermal behavior within an extremely constrained physical volume, as traditional cooling methods like large heatsinks are often impractical, necessitating advanced control strategies to improve efficiency and limit heat generation.

The three-phase six-switch MOSFET voltage source inverter is the most commonly used industrial inverter structure, and researchers are widely exploring how to enhance inverter efficiency by modifying pulse width modulation methods. Regarding traditional continuous pulse width modulation (CPWM), each active power switch comprises one turn-on and one turn-off transition within one carrier period, which often leads to excessive switching losses and thermal issues in compact systems. Discontinuous pulse width modulation (DPWM), compared to CPWM, can reduce switching frequency and improve inverter efficiency, making it widely applicable in motor control [6,7], photovoltaic inverters [8], and controlled rectifiers [9,10].

Classical DPWM can be divided into six modes according to the range of the bus voltage clamping area: 120° DC bus voltage clamping DPWM modes (DPWMmax and DPWMmin), 60° DC bus voltage clamping DPWM modes (DPWM0, DPWM1, and DPWM2), and a 30° DC bus voltage clamping DPWM mode (DPWM3) [11,12,13,14]. There are two commonly used methods for implementing discontinuous DPWM strategies. The first method is to inject zero-sequence voltage into a three-phase sinusoidal modulation waveform [7,8,10], and the other is to change the zero vector interval in a segment of the timing diagram [12]. As discussed in [15,16], compared with traditional CPWM, the modulation wave of DPWM is clamped to the positive or negative DC bus voltage. During this period, the inverter switches do not operate, reducing the number of switching times. Therefore, the switching loss of the six classic DPWM schemes is lower than that of the CPWM scheme, thereby improving the efficiency of the inverter.

Regarding the harmonic characteristics of DPWM, studies [11,14,17] analyze the harmonic distortion rate function to characterize DPWM’s harmonic properties and explore the relationship between the modulation index (m) and harmonic distortion. Meanwhile, reference [18] investigates harmonic characteristics by analyzing inductors’ charge–discharge behavior with different PWM patterns. Some other scholars have attempted to propose unified modulation methods for the six DPWM modes [13,14,15]. For example, reference [15] selects phase shifts of −30°, 0°, and 30° as the switching conditions for DPWM modes, while references [19,20] use the power factor angle as the switching condition, achieving a 50% reduction in switching losses within the power factor range of [−30°, 30°]. There are also other DPWM methods, as studied in [21], which extend from a single-objective optimization problem to multi-objective optimization, including linear modulation range expansion, common-mode voltage reduction [22], current ripple, and switching losses [23]. References [24,25] propose a split DC bus voltage clamping DPWM and achieve its control effect by altering the sequence of zero-voltage vector actions, such as “0127”, “012”, “721”, “0121”, or “7212.”

In summary, factors such as frequency, current, voltage, and power factor in motor control can vary according to operating conditions. Current discussions on studies mostly focus on a fixed power factor or a small range of power factor angles [−60°, 60°]. However, under light-load or low-speed conditions, the power factor angle can exceed 60°, and traditional DPWM modes fail to achieve the lowest losses in low power factor ranges. Therefore, it is essential to study optimal DPWM methods for a wide power factor range.

The question of how to obtain accurate power factor is another hot research topic for scholars. The measurement of power factor under non-sinusoidal conditions has significantly evolved in the power electronics literature, driven by increasing harmonic distortion in inverter drives and grid application. Traditional methods relying on zero-crossing detection or single-point phase-angle calculations (e.g., [26]) fail under harmonic distortion conditions as they cannot isolate fundamental components. This limitation spurred the adoption of frequency-domain approaches, such as Fast Fourier Transform (FFT) [27,28], which provide accurate harmonic decomposition but suffer from computational complexity and latency, making them unsuitable for real-time control. Recent advancements proposed time-domain techniques with adaptive filtering and phase-locked loops (PLLs) [27,28,29]. The Synchronous Reference Frame PLL (SRF-PLL) [26] has emerged as a gold standard for fundamental component extraction, enabling precise displacement power factor (DPF) calculation via phase angle difference. For the distortion power factor, the literature converges on RMS-based formulations [9] quantifying harmonic energy separation.

Targeting the full power factor domain [−90°, 90°], this paper proposes a phase-shift-angle-based discontinuous PWM modulation method for voltage source inverters. This algorithm directly injects zero-voltage vectors using three-phase reference voltages and introduces a phase shift angle to achieve efficiency-optimal control across the full power factor range of [−90°, 90°]. The power factor is obtained through the Synchronous Reference Frame Phase-Locked Loop (SRF-PLL) method, which can be applied in the current harmonics situation. In addition to including the six classic DPWM modes, this method extends to split DC bus voltage clamping DPWM modes under low-power-factor conditions. This paper also analyzes inverter switching losses and current harmonics under arbitrary power factor angles. Simulations and experiments verify the high efficiency, low loss, and rapid response of the proposed DPWM method with adjustable phase shift angles.

## 2. Discontinuous PWM Modulation Method with Phase Shift Angle

### 2.1. Basic Principle of Zero-Sequence Voltage Injection

A typical 3ph VSI system consists of a 24V DC bus voltage from a battery, a MOSFET-based inverter, and a star-connected inductive-resistive load (L, R) or motor load, as shown in Figure 1. The inverter comprises six switching devices, labeled VT1 to VT6. The upper switch, VT1, and the lower switch, VT4, are controlled by complementary gate signals S_a_ and $\bar{S_{a}}\;$to regulate the A-phase voltage or current. Similarly, the upper switch, VT3, and the lower switch, VT6, are controlled by complementary gate signals S_b_ and $\bar{S_{b}}\;$for the B-phase voltage or current, while the upper switch, VT2, and the lower switch, VT5, are controlled by complementary gate signals S_c_ and $\bar{S_{c}}$ for the C-phase voltage or current.

The three-phase target voltage modulation waves, $V_{a},V_{b},V_{c}$, which are 120° apart from each other, are expressed as(1)$$\left\{\begin{array}{c} V_{a}=\mathrm{Mcos}(\theta_{e})\\ V_{b}=\mathrm{Mcos}(\theta_{e}-\frac{2}{3}\pi )\\ V_{c}=\mathrm{Mcos}(\theta_{e}+\frac{2}{3}\pi ) \end{array}\right.$$
where M is the amplitude of the modulation wave. By comparing the triangular carrier wave with the A-phase reference voltage waveform, the gate signal, S_a_, for the upper switch, VT1, can be obtained, as illustrated in Figure 2a. The gate drive signals for the B- and C-phase switches (S_b_ and S_c_) can be derived in the same manner. The offset voltage can be added to the three modulation voltages in (2), as shown in Figure 2b.(2)$$\left\{\begin{array}{c} V_{a}^{*}=V_{a}+V_{z}\\ V_{b}^{*}=V_{b}+V_{z}\\ V_{c}^{*}=V_{c}+V_{z} \end{array}\right.$$
where $V_{a}^{*},V_{b}^{*},V_{c}^{*}$ represent the three-phase voltage modulation waves after the injection of the zero-sequence voltage.

Since the line current sinusoidal waveform serves as the control target, the injected zero-sequence offset voltage has no impact on the final current waveform (since $V_{ab}=V_{a}-V_{b}=V_{a}^{*}-V_{b}^{*}$). The maximum–minimum functions are defined in (3), respectively, as(3)$$\left\{\begin{array}{c} V_{max}={max(V}_{a},V_{b},V_{c})\\ V_{min}={min(V}_{a},V_{b},V_{c}) \end{array}\right.$$

Due to the DC bus voltage constraints of the 3ph-VSI, the zero-sequence voltage can be calculated as(4)$$V_{z}=-kV_{max}-\left(1-k\right)V_{min}+2k-1$$
where k is the zero-sequence voltage selection coefficient, as described in reference [15]. Figure 3 illustrates PWM modes under different k values, where the blue line represents the A-phase modulation wave, the yellow line denotes the injected zero-sequence voltage, and the red line shows the modified A-phase modulation wave after zero-sequence voltage injection.

Specifically, when k = 0.5, this corresponds to the classic SVPWM mode. As discussed in references [11,12,13,14,15], DPWMmax (k = 1) and DPWMmin (k = 0) exhibit 120° DC bus clamping DPWM mode.

### 2.2. Zero-Sequence Voltage Generation Method with Phase Shift Angle

Unlike the approaches in references [11,12,15,16,19], this paper proposes the DPWM method that incorporates a phase shift angle into the zero-sequence voltage generation, as illustrated in Figure 4.

The specific implementation of Figure 4 is as follows:

Step 1. Perform the Clarke transformation from the three-phase modulation voltage waves $V_{a},V_{b},V_{c}$ into $V_{\alpha }$ and $V_{\beta }$:(5)$$\left[\begin{array}{c} V_{\alpha }\\ V_{\beta } \end{array}\right]=\frac{2}{3}\left[\begin{array}{ccc} 1 & -\frac{1}{2} & -\frac{1}{2}\\ 0 & \frac{\sqrt{3}}{2} & -\frac{\sqrt{3}}{2} \end{array}\right]\left[\begin{array}{c} V_{a}\\ V_{b}\\ V_{c} \end{array}\right]$$

Step 2. Obtain the angle, θ_e_, of the resultant voltage vector using the arc tangent function, as shown in Equation (6):(6)$$\theta_{e}=arctangent(\frac{V_{\beta }}{V_{\alpha }})$$

Step 3. Introduce a phase shift angle φ superimposed on θ_e_. Then, by applying triple-frequency operation and modulo function (Equation (7)), we obtain the conditional phase φ_s_(7)$$\varphi_{s}=rem[{3(\theta }_{e}+\varphi +2\pi ),2\pi ]$$

Step 4. The rem function denotes the modulo operation with respect to 2π, and k can be selected based on the created conditional phase φ_s_ in (8)(8)$$k=\left\{\begin{array}{c} 1\;\;\;\;\;\;\;\;\;\varphi_{s}\geq \pi \;\\ 0\;\;\;\;\;\;\;\;\;\varphi_{s}<\pi \end{array}\right.$$

Step 5. Due to the DC bus voltage constraints of the 3ph-VSI, the zero-sequence voltage $V_{z}$ can be calculated using (4), and the final modulated voltage waveforms can be calculated using (2).

The approach proposed in this paper represents a more comprehensive DC bus clamping DPWM strategy that extends beyond the conventional DPWM1 and DPWM2 modes described in [7].

For example:

When φ = π/2 (90°), the system operates in the traditional DPWM1 mode, as illustrated in Figure 5a.

When φ = π/3 (60°), the system switches to DPWM2 mode, as shown in Figure 5b.

When φ is set to be 5π/12 (75°), which is continuously located within π/3 to π/2 (60–90°), and the 60° clamping interval shifts smoothly, intermediate modulation patterns are generated between those of Figure 5a,b, as shown in Figure 5c.

Furthermore, this method can additionally construct 60° DC bus clamping PWM modes. For instance:

When φ = π/6 (30°) (Figure 5d), the system operates in the conventional DPWM3 mode.

When φ = π/5 (36°) (Figure 5e), a 24°/36° split DC bus clamping mode (DPWM-24°/36°) is realized.

When φ = π/4 (45°) (Figure 5f), a 15°/45° split DC bus clamping mode (DPWM-15°/45°) is achieved.

Once inverter parameters are fixed, its losses are primarily governed by two factors: the number of switching actions per period and the current magnitude at the moment of turn-on. The power factor angle, which defines the phase relationship between the modulation voltage and current, is also critical. While continuous PWM (CPWM) has a switching frequency equal to its carrier frequency, discontinuous PWM (DPWM) suspends switching for intervals, resulting in a lower effective frequency. For instance, the six classic DPWM modes can reduce the switching frequency by one-third. Losses are further minimized if switching is avoided during peak current intervals.

This study examines arbitrary loads across the entire power factor range of [-π/2, π/2], including both inductive and capacitive types. For inductive loads, the analysis focuses on switching losses and current harmonic characteristics under DPWM1, DPWM2, DPWM3, and split DC bus clamping modes. The behavior for capacitive loads can be derived by symmetry.

## 3. Analysis of Switching Characteristics and Harmonic Performance for All DC Bus-Clamping DPWM Modes

### 3.1. Analysis of DPWM Switching Characteristics

As illustrated in Figure 6, A-phase bridge leg switches are analyzed under an inductive load. When the upper switch VT1 turn-on signal, S_a_, equals 1, the phase A drain current $i_{D}$ rises from zero to its peak value. The source-drain voltage $V_{DS}$ of the upper switch drops from $V_{DS}$ to 0. When the upper switch VT1 turn-off signal, S_a_, equals 0, the current $i_{D}$ gradually decreases from the peak to zero. The source-drain voltage of the upper switch rises back to $V_{DS}$.

The area of the switching losses is enclosed by the voltage and current area. The single turn-on loss and turn-off loss can be approximated as(9)$$P_{\mathrm{sw}}=\frac{1}{2}V_{DS}i_{D}t_{on}+\frac{1}{2}V_{DS}i_{D}t_{off}$$

In the above equation, t_on_ and t_off_ represent the on and off times of the power switch, respectively. If CPWM is implemented on the bridge arm, the switching losses can be expressed in (10) within the current cycle for the phase A current.(10)$$P_{\mathrm{sw}\text{\_}\mathrm{CPWM}}=\frac{1}{2\pi }\frac{V_{dc}\left(t_{on}+t_{off}\right)}{2T_{s}}\int_{-\pi }^{\pi }f_{i}(\theta_{e})d\theta_{e}$$

If the current is a sinusoidal current waveform, the power factor $\varphi_{pf}$ is defined as the angle between the voltage and current, as shown in (11):(11)$$f_{i}(\theta_{e})=\left|i_{a}\right|=I_{max}\left|cos(\theta_{e}-\varphi_{pf})\right|$$

Since the power factor is independent of integration in (10), the final losses $P_{\mathrm{sw}\text{\_}\mathrm{CPWM}}$ for phase A can be expressed as(12)$$P_{\mathrm{sw}\text{\_}\mathrm{CPWM}}=\frac{V_{dc}I_{max}}{\pi T_{s}}\left(t_{on}+t_{off}\right)$$

However, if DPWM is used, the presence of the bus voltage clamping interval significantly impacts switching losses. For example, in DPWM2, within one electrical cycle, there are two 60-degree intervals without switching, which reduces switching losses. Figure 7 depicts the current and modulation voltage waveform under DPWM2, assuming a 45° lagging power factor angle.

In the [0, 60°] interval, there is no switching for phase A, the switching loss is zero, and the current function $f_{i}\left(\theta_{e}\right)$ can be changed to (13).(13)$$f_{i}\left(\theta_{e}\right)=\left\{\begin{array}{c} 0\;\;\;\;\;\;\;\;\;\;\;\;\;\left|V_{a}^{*}\right|\geq \frac{V_{dc}}{2}\\ \left|i_{a}\right|\;\;\;\;\;\;\;\;\;\;\;\;\left|V_{a}^{*}\right|<\frac{V_{dc}}{2} \end{array}\right.$$

Since the voltage and current waveform have half-wave symmetry, the average switching loss of the fundamental wave of the phase A current in one cycle can be obtained by integrating the half cycle [−90°, 90°] as (14).(14)$$P_{\mathrm{sw}\text{\_}\mathrm{DPWM}2}(\varphi_{pf})=\frac{1}{2\pi }\frac{V_{dc}\left(t_{on}+t_{off}\right)}{2T_{s}}{2I}_{max}(\int_{-\frac{\pi }{2}}^{0}\left|\cos\left(\theta_{e}-\varphi_{pf}\right)\right|d\theta_{e}\;+\int_{\frac{\pi }{3}}^{\frac{\pi }{2}}\left|\cos\left(\theta_{e}-\varphi_{pf}\right)\right|d\theta_{e})$$

Define the switching loss ratio function as SLRF, which is the ratio of the switching loss function of all DC bus clamping DPWM modes to the switching loss function of CPWM:(15)$$SLRF=\frac{P_{\mathrm{sw}\text{\_}\mathrm{DPWM}2}(\varphi_{pf})}{P_{\mathrm{sw}\text{\_}\mathrm{CPWM}}}$$

Based on the above analysis, SLFR can be used to describe the reduction in switching losses under different DPWM modes compared to CPWM. Figure 8 plots the SLRF of DPWM0-3 as a function of the power factor angle. The green curve represents DPWM0, the blue curve represents DPWM1, the red curve represents DPWM2, and the cyan curve represents DPWM3. Taking the red curve (DPWM2) as an example, as the power factor angle changes from −90° to 90°, the blue curve shows a decreasing trend, reaches a minimum, and then rises again. The black curve, which covers all DPWM modes, represents the optimal DPWM mode with the lowest SLRF.

As analyzed from Figure 8, the DPWM with the lowest switching loss can be categorized as follows according to the power factor $\varphi_{pf}$:

Mode 1: When $\varphi_{pf}$ = 0°, the voltage and current are in phase, and the DPWM1 mode (marked A in Figure 8) has the lowest switching loss. The switching loss is half that of SVPWM and SPWM. The corresponding phase shift angle φ is 90°.

Mode 2: When ${0{}^{\circ}<\varphi }_{pf}$< 30°, the midpoint of the 60° clamping region is aligned with the current peak, automatically adjusting with the power factor changes. The intermediate mode is between DPWM1 and DPWM2 (marked B in Figure 8). The switching loss is half that of SVPWM and SPWM. The corresponding phase shift angle φ is [60°, 90°].

Mode 3: When ${30{}^{\circ}<\varphi }_{pf}$< 60°, the DPWM2 mode (marked C in Figure 8) has the lowest switching loss. The switching loss is 0.5–0.567 times that of SVPWM and SPWM. The corresponding phase shift angle φ is 60°.

Mode 4: When ${60{}^{\circ}\leq \varphi }_{pf}$< 90°, the lowest switching loss generates the split DC bus clamping DPWM mode (marked D in Figure 8). Based on the power factor, the clamping range is automatically switched. In this case, the switching loss is 0.567–0.634 times that of SVPWM and SPWM. The corresponding phase shift angle φ range is [30°, 60°].

Mode 5: $\varphi_{pf}$ = 90°, generating the DPWM3 mode (marked E in Figure 8). The switching loss is 0.634 times that of SVPWM. The corresponding phase shift angle φ is 30°.

By varying the phase shift angle φ, the minimum switching losses DPWM can be achieved across the entire power factor range of [−90°, 90°]. The corresponding relationship is



(16)
$$\left\{\begin{array}{c} \varphi =\frac{\pi }{3}-\varphi_{pf}\;\;\;\;\;\;-\frac{\pi }{2}\leq \varphi_{pf}<-\frac{\pi }{3}\\ \varphi =\frac{2\pi }{3}\;\;\;\;\;\;\;\;\;\;\;\;\;\;\;-\frac{\pi }{3}\leq \varphi_{pf}<-\frac{\pi }{6}\\ \varphi =\frac{\pi }{2}-\varphi_{pf}\;\;\;\;\;\;\;\;\;-\frac{\pi }{6}\leq \varphi_{pf}<\frac{\pi }{6}\\ \varphi =\frac{\pi }{3}\;\;\;\;\;\;\;\;\;\;\;\;\;\;\;\;\;\;\;\;\;\;\;\;\;\;\frac{\pi }{6}\leq \varphi_{pf}\leq \frac{\pi }{3}\\ {\varphi =\frac{2\pi }{3}-\varphi }_{pf}\;\;\;\;\;\;\;\;\;\;\;\;\frac{\pi }{3}\leq \varphi_{pf}\leq \frac{\pi }{2} \end{array}\right.$$



The relationship of the phase shift angle in (16) and the power factor angle is depicted in Figure 9, where the key turning points are marked with red dots. In the entire power factor range [−90°, 90°], different phase shift angles φ can be obtained according to different power factors $\varphi_{pf}$, thereby obtaining the DPWM mode with the minimum SLRF. Compared with references [10,11,16], the lowest loss DPWM at the power factor angles of [−90°, −60°] and [60°, 90°], namely split DC bus clamping DPWM mode, is added, which expands the full power factor range.

### 3.2. Real-Time Power Factor Extraction Using SRF-PLL

To accurately extract the real-time power factor component under harmonic current distortion, the SRF-PLL method was proposed. The three-phase currents/voltages are first transformed into the stationary αβ-frame via the Clarke transform and then rotated into the synchronous dq-frame using an angle estimated using a PLL. The PLL regulates the q-axis current/voltage to zero, ensuring alignment with the fundamental frequency, while the d-axis component contains the fundamental magnitude after low-pass filtering. The control block is shown in Figure 10.

The fifth and seventh harmonics are used for further explanation, since they are common seen from deadtime effect of VSI. A three-phase current or voltage signal with fifth and seventh harmonics can be written as(17)$$\left[\begin{array}{c} i_{a}\\ i_{b}\\ i_{c} \end{array}\right]=I_{1}\left[\begin{array}{c} \sin(\theta_{i})\\ \sin(\theta_{i}-\frac{2}{3}\pi )\\ \sin(\theta_{i}+\frac{2}{3}\pi ) \end{array}\right]+I_{5}\left[\begin{array}{c} \sin(-5\theta_{i}+\varphi_{5})\\ \sin[-5(\theta_{i}-\frac{2}{3}\pi )+\varphi_{5}]\\ \sin[-5(\theta_{i}+\frac{2}{3}\pi )+\varphi_{5}] \end{array}\right]+I_{7}\left[\begin{array}{c} \sin(7\theta_{i}+\varphi_{7})\\ \sin\;[7(\theta_{i}-\frac{2}{3}\pi )+\varphi_{7}]\\ \sin\;[7(\theta_{i}+\frac{2}{3}\pi )+\varphi_{7}] \end{array}\right]$$

After Clark and park transformation,(18)$$\left[\begin{array}{c} i_{d}\\ i_{q} \end{array}\right]=\left[\begin{array}{cc} cos{\hat{\theta }}_{i} & sin{\hat{\theta }}_{i}\\ -sin{\hat{\theta }}_{i} & cos{\hat{\theta }}_{i} \end{array}\right]\frac{2}{3}\left[\begin{array}{ccc} 1 & -\frac{1}{2} & -\frac{1}{2}\\ 0 & \frac{\sqrt{3}}{2} & -\frac{\sqrt{3}}{2} \end{array}\right]\left[\begin{array}{c} i_{a}\\ i_{b}\\ i_{c} \end{array}\right]$$

The angle for park transformation is obtained from the PLL estimation value ${\hat{\theta }}_{i}$. The total current in the dq frame is the sum of the fundamental DC component and the AC components from the harmonics:(19)$$\left[\begin{array}{c} i_{d}\\ i_{q} \end{array}\right]\approx I_{1}\left[\begin{array}{c} \cos{(\theta }_{i}-{\hat{\theta }}_{i})\\ \sin{(\theta }_{i}-{\hat{\theta }}_{i}) \end{array}\right]+I_{5}\left[\begin{array}{c} \cos(6\theta_{i}+\varphi_{5})\\ \sin(6\theta_{i}+\varphi_{5}) \end{array}\right]+I_{7}\left[\begin{array}{c} \cos(6\theta_{i}+\varphi_{7})\\ \sin(6\theta_{i}+\varphi_{7}) \end{array}\right]$$
where $I_{1}cos{(\theta }_{i}-{\hat{\theta }}_{i})$ and $I_{1}sin{(\theta }_{i}-{\hat{\theta }}_{i})$ are the desired DC components representing the amplitude and phase of the fundamental positive-sequence current.

The sixth harmonic ripple on iq will be superimposed on this error signal. The terms oscillating at $6\theta_{i}$ are the undesired AC ripple caused by the fifth and seventh harmonics and can be filtered by low-pass filters or suppressed by a suitable bandwidth from the PLL. The PLL uses the q-axis component (iq for an SRF-PLL) as its error signal.

When the estimation angle error $\theta_{i}-{\hat{\theta }}_{i}$ is close to zero, the PLL relationship can be approximated to be sin${(\theta }_{i}-{\hat{\theta }}_{i}$) ≈ ${(\theta }_{i}-{\hat{\theta }}_{i}$), the open loop of the PLL can be expressed as$$(K_{p}+K_{i}\frac{1}{s})\frac{1}{s}(\theta_{i}-{\hat{\theta }}_{i})={\hat{\theta }}_{i}$$

The closed-loop PLL can be expressed as$$\frac{{\hat{\theta }}_{i}}{\theta_{i}}=\frac{K_{p}s+K_{i}}{{s^{2}+K}_{p}s+K_{i}}$$

The closed-loop PLL is a typical second order system, and the second-order system can be used to ensure stability.

This approach effectively isolates the fundamental component, even under high harmonic distortion, making it suitable for applications such as grid-tied inverters and motor drives. The method provides real-time adaptability to frequency variations and superior harmonic rejection compared to fixed-filter techniques. The simulation results are shown in Figure 11. The voltage is a blue waveform with a 5% fifth harmonic, and the current is a red waveform with a 20% fifth harmonic and the 15% seventh harmonic. The frequency spectrum is shown in Figure 11c. Since the current harmonics will produce zero-crossing offset, the total power factor will be higher than the fundamental power factor (also known as displacement power factor in Figure 11b).

### 3.3. Harmonic Characteristics Analysis

Ref. [17] defines the harmonic distortion factor (HDF) to describe the square RMS value of the harmonic current by applying different modulation modes. The HDF of SVPWM and DPWM1–3 is(20)$$\left\{\begin{array}{c} {HDF}_{SVPWM}=\frac{3}{2}m^{2}-\frac{4\sqrt{3}}{\pi }m^{3}+(\frac{27}{16}-\frac{81\sqrt{3}}{64\pi })m^{4}\\ {HDF}_{DPWM1}=6m^{2}-\frac{8\sqrt{3}+45}{2\pi }m^{3}+(\frac{27}{8}+\frac{27\sqrt{3}}{32\pi })m^{4}\\ \begin{array}{c} {HDF}_{DPWM2}=6m^{2}-\frac{35\sqrt{3}}{2\pi }m^{3}+\left(\frac{27}{8}+\frac{81\sqrt{3}}{64\pi }\right)m^{4}\\ {HDF}_{DPWM3}=6m^{2}-\frac{-62\sqrt{3}+45}{2\pi }m^{3}+(\frac{27}{8}-\frac{27\sqrt{3}}{16\pi })m^{4} \end{array} \end{array}\right.$$
where m is the modulation index, which can be defined as(21)$$m=\frac{\sqrt{V_{\alpha }^{2}+V_{\beta }^{2}}}{\frac{V_{dc}}{2}}$$

When selecting SPWM mode, the maximum modulation index is 1. If SVPWM or DPWM is selected, the maximum modulation index can be improved to be 1.15 with third harmonics injection. If the 3ph-VSI undergoes the six-step operation, the modulation index m can reach 1.265 times from fundamental voltage, however, there is higher distortion in the six-step operating mode.

As shown in (20), the HDF function is a single-valued function of the modulation index, m. Figure 12a,b show the HDF functions at different switching frequencies. Since DPWM can reduce the switching frequency by 1/3 compared to SVPWM, the carrier frequency can be increased by approximately 3/2 while maintaining the same switching frequency compared to SVPWM. Therefore, the HDF of the DPWM is multiplied by 3/2 in Formula (20), resulting in Figure 12b. The split bus clamping DPWM mode is located between DPWM2 and DPWM3, represented by the shaded area in Figure 12.

As shown in Figure 12a, at the same switching frequency, all DPWM modes have a higher harmonic content than SVPWM. However, when the modulation index is below 0.2 or above 1 in the high modulation index region, DPWM exhibits a comparable harmonic content to SVPWM. When the switching frequency increases to 1.5 times of that in Figure 12a, the HDF function can be shown in Figure 12b. When the modulation index is above approximately 0.8, DPWM exhibits lower harmonics than SVPWM.

## 4. Experimental Results

To verify the efficiency improvement and harmonic characteristics of the proposed DPWM control strategy with the phase shift angle, four experiments were conducted as follows: an inverter efficiency test, a power factor adaptation test, a power factor dynamic test, and a harmonics evaluation test. A 3ph-VSI circuit and resistive–inductive load experimental test bench are shown in Figure 13a. The 3ph-VSI is a typical onboard micro inverter supplied by the DC battery 24 V. The inverter switching frequency is 20 kHz, and the HY3610P as six N-channel MOSFET power switches is used with an internal resistance of 4.5 mΩ. The power of the inverter is 120 W. The series load resistance and inductance are 1 Ω–5 Ω and 2 mH, respectively. The maximum current can reach 5A-RMS. The main control chip uses a digital signal processor (DSP, type: TMS320F28335PGFA, Texas Instruments, Dallas, TX, USA) with a dead time of 2.5 µs. The DSP uses a serial peripheral interface (SPI) to transfer intermediate variables to the 16-bit digital-to-analog (DA) peripheral, which can be monitored by an oscilloscope. The total harmonic distortion (THD), voltage, current, power angle, and harmonic information of the three-phase load are obtained using a PA4400A power analyzer (Powertek Ltd., Berkshire, UK).

The systematic control block can be shown in Figure 13b. As analyzed above, the system control module includes the above-mentioned SRF-PLL voltage and current sub-block, zero-sequence voltage generation sub-block, and gate signal generator sub-block.

### 4.1. Inverter Efficiency Test

To evaluate inverter efficiency and power losses, the output current was controlled to a constant of 3 A with a fundamental frequency of 60 Hz. The voltage and current of the input RL circuit (1 Ω, 2 mH) were measured using a PA4400A power analyzer. The resulting efficiency under different discontinuous PWM (DPWM) schemes is shown in Figure 14a.

With a frequency of 60 Hz, the power factor angle was approximately 37°. As predicted by the analysis in Section 2.1, DPWM2 achieved the highest efficiency. The experimental results confirm this, showing an efficiency of 67.81% for DPWM2 compared to 64.72% for SVPWM—an improvement of 3.1%. Seen from Table 1 and Table 2, the control chip and peripheral circuits consumed 3.74 W, measured during power-on. Furthermore, the DPWM2 inverter’s combined switching and conduction losses were approximately 15.44 W, a 15% reduction from the 18.16 W lost with SVPWM.

A separate test was conducted at 400 Hz with the current maintained at 1.5 A, the results of which are shown in Figure 14b. Here, the power factor angle increased to approximately 78°. At this higher frequency, the split DC bus clamping DPWM mode demonstrated the highest efficiency, as anticipated. Compared to SVPWM, this method improved efficiency by 6.14% and reduced losses by 39.4% (from 3.09 W to 1.87 W), which aligns with the theoretical analysis. Detailed data are provided in Table 1 and Table 2, with further quantitative analysis available in the literature [29,30].

### 4.2. Testing for Different Modulation Strategies with Different Power Factors

To validate the impact of different discontinuous PWM (DPWM) modes on power factor adaptation, an experimental study was conducted under varying load and frequency conditions.

The experiment began with a modulation frequency of 21 Hz, using an inductor (L) of 2 mH and a resistor (R) of 1 Ω, resulting in a power factor angle of approximately 15°. Waveforms for the modulation voltage, zero-sequence injected voltage, current, and output voltage were captured using a digital-to-analog converter (DA), current probes, and differential voltage probes, respectively, as shown in the oscilloscope image in Figure 15a. The results demonstrate a voltage clamping scheme hybridizing DPWM1 and DPWM2, where the midpoint of the 60° clamping range was aligned with the current peak. This alignment kept the A-phase bridge arm switch clamped at the current peak, significantly reducing switching losses.

When the modulation frequency was increased to 46 Hz (with L = 2 mH and R = 2 Ω, yielding a power factor angle of ~30°), the system transitioned from DPWM1 to DPWM2 mode, as shown in Figure 15b. This transition confirmed that the clamping voltage varies with the power factor angle within the 0–60° range.

At a further increased frequency of 80 Hz (power factor angle ~45°), the DPWM2 mode dynamically adapted to the changing power factor, as illustrated in Figure 15c. Raising the frequency to 221 Hz (power factor angle ~72°) necessitated a switch to a split DC bus clamping DPWM mode. In this mode, the 60° voltage interval was divided into two segments—12° and 48°—for clamping during the first half of the current cycle, resulting in two distinct clamping intervals for the bridge arm (Figure 15d). Finally, at 424 Hz (power factor angle ~80°), the DPWM scheme split the 60° clamping interval into 20° and 40° segments, as presented in Figure 15e.

These results demonstrate that the optimal selection of DPWM modes and their clamping intervals is highly dependent on the power factor angle, which itself varies with modulation frequency and load. The findings confirm the effectiveness of adaptive DPWM strategies in minimizing switching losses while maintaining desired performance.

To verify the dynamic performance of the DPWM mode with power factor angle variation and sudden changes, the following experiment was designed. With a given modulation frequency of 50 Hz, the given frequency suddenly increased to 300 Hz. The experimental results of the modulation voltage, zero-sequence injected voltage, switching signals, and current waveforms are shown in Figure 16a. Given a modulation frequency of 300 Hz that suddenly drops to 80 Hz, the experimental results are shown in Figure 16b.

As shown in Figure 16a, when the modulation frequency is 50 Hz, the power factor angle is approximately 32°, indicating DPWM2 mode. However, due to a sudden change in frequency to 300 Hz, the power factor drops to 75°, entering split DC bus clamping DPWM mode. As shown in Figure 16b, after the modulation frequency drops to 80 Hz, the power factor angle drops to approximately 45°, returning to DPWM2 mode. The proposed DPWM strategy can maintain good dynamic performance under sudden power factor changes and switches to DPWM mode for minimal switching losses.

### 4.3. Testing for Harmonics

In order to verify the harmonic characteristics shown in Section 3.2 and the current harmonic characteristics under high modulation conditions, the following experiment is designed. Given a modulation frequency of 50 Hz and an adjustment voltage amplitude of 12 V, the inverter enters the high modulation area. In this case, the current of about 3.6 A can be obtained through the power analyzer. The SVPWM mode with a carrier frequency of 12 kHz, the DPWM2 mode with a carrier frequency of 12 kHz, and the DPWM2 mode with a carrier frequency of 20 kHz are compared, respectively, as shown in Figure 17a–c.

When comparing the experimental results shown in Figure 17a,b, under the given frequency of 50 Hz and the given modulation voltage amplitude of 12 V, it can be seen that the high modulation index mode is entered, and DPWM2 mode is preferably entered. The main harmonic orders are the fifth and seventh harmonics. At a switching frequency of 12 kHz, the current THD in SVPWM mode is 2.25%, and the current THD in DPWM2 mode is 2.69%. The CPWM and DPWM2 modes are comparable, with the DPWM2 current distortion being slightly higher. When comparing the experimental results shown in Figure 17a,c, it can be seen that by increasing the switching frequency to 20 kHz, the THD of DPWM can be reduced to 1.84% compared to SVPWM.

## 5. Conclusions

This paper presents a novel minimum-switching-loss discontinuous PWM (DPWM) method with all DC bus clamping DPWM modes used in a 120 W onboard micro-inverter, enabled by an adaptive phase-shift angle that adjusts according to the power factor. By dynamically generating optimal DPWM strategies based on the power factor angle, the proposed approach extends conventional six-segment DPWM modes into unlimited adaptive configurations, ensuring minimized switching losses across all operating conditions.

Key contributions and performance advantages include the following:(1).Switching Loss Reduction: It maintains 40% lower switching losses compared with CPWM even under extremely low-power-factor conditions, significantly improving inverter efficiency over the full operating range.(2).Improved Waveform Quality: It reduces the current THD from 2.25% to 1.84% at high modulation indices compared to traditional SVPWM while preserving equivalent switching losses.(3).Robust Dynamic Performance: It demonstrates seamless adaptation to power factor transients and extreme operational conditions, ensuring stable and efficient operation.

The proposed method not only enhances energy efficiency but also provides a flexible and scalable solution for modern power electronic systems, making it particularly suitable for applications requiring wide power factor variations and high efficiency demands.

## Figures and Tables

**Figure 1 micromachines-16-01128-f001:**
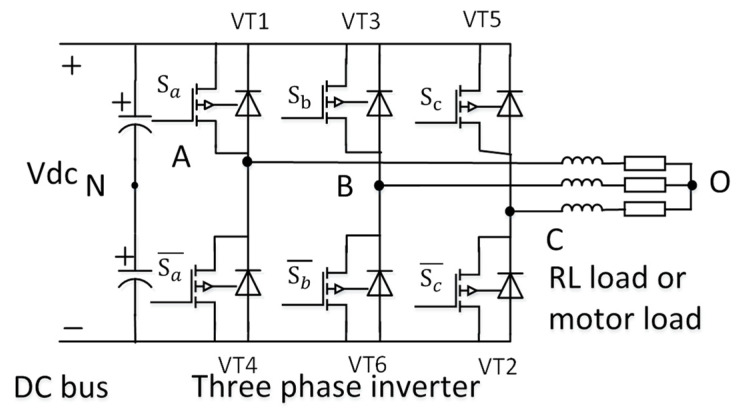
Three-phase voltage source inverter system.

**Figure 2 micromachines-16-01128-f002:**
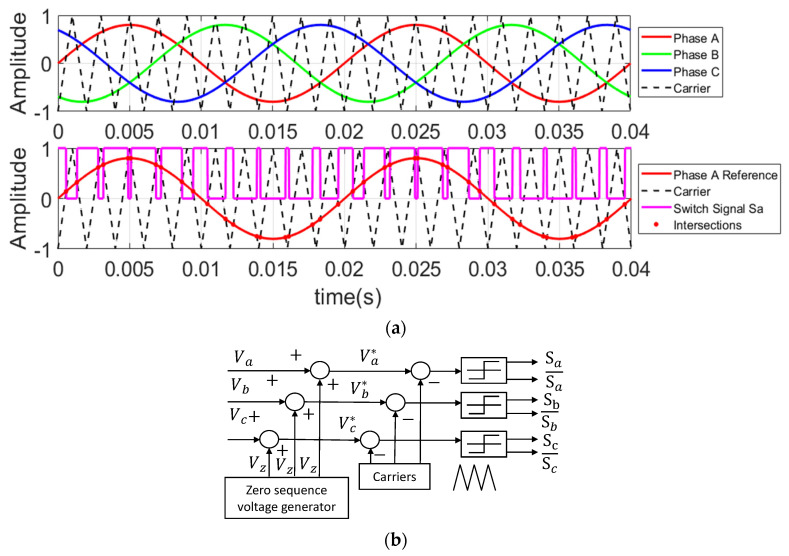
The relationship between injection voltage, carriers, and modulated voltage. (**a**) Gate signal generation; (**b**) the zero-sequence voltage injection.

**Figure 3 micromachines-16-01128-f003:**
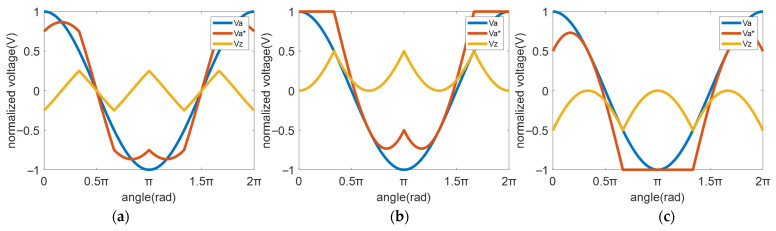
The modulation waveform with zero-sequence injection. (**a**) SVPWM, k = 0.5; (**b**) DPWMmax, k = 1; (**c**) DPWMmin, k = 0.

**Figure 4 micromachines-16-01128-f004:**
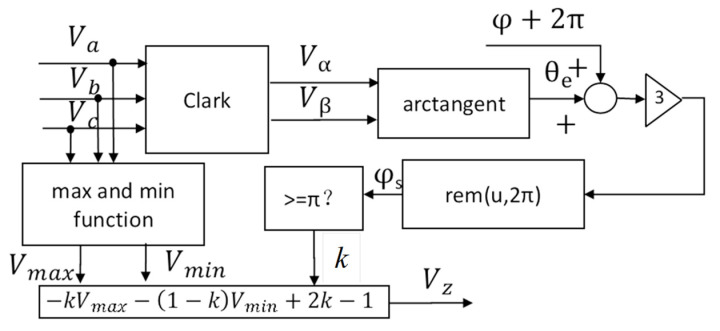
The proposed zero-sequence voltage generator with the phase shift angle *φ*.

**Figure 5 micromachines-16-01128-f005:**
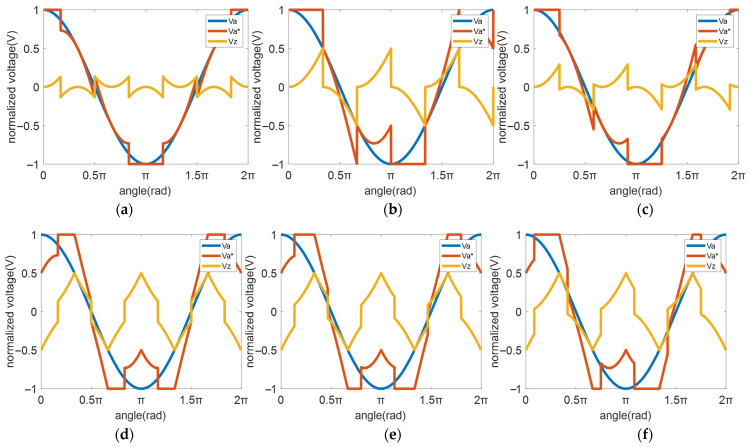
PWM strategies with different φ values: (**a**) φ = π/2, DPWM1. (**b**) φ = π/3, DPWM2. (**c**). φ = 5π/12, DPWM1–2. (**d**) φ = π/6, DPWM3. (**e**) φ = π/5, DPWM24°/36°. (**f**) φ = π/4, DPWM15°/45°.

**Figure 6 micromachines-16-01128-f006:**
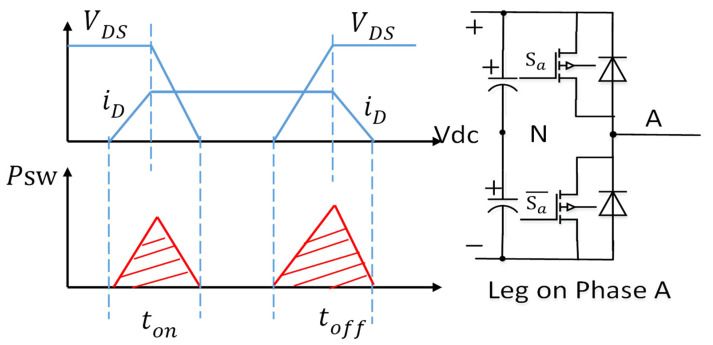
Analysis of switching losses.

**Figure 7 micromachines-16-01128-f007:**
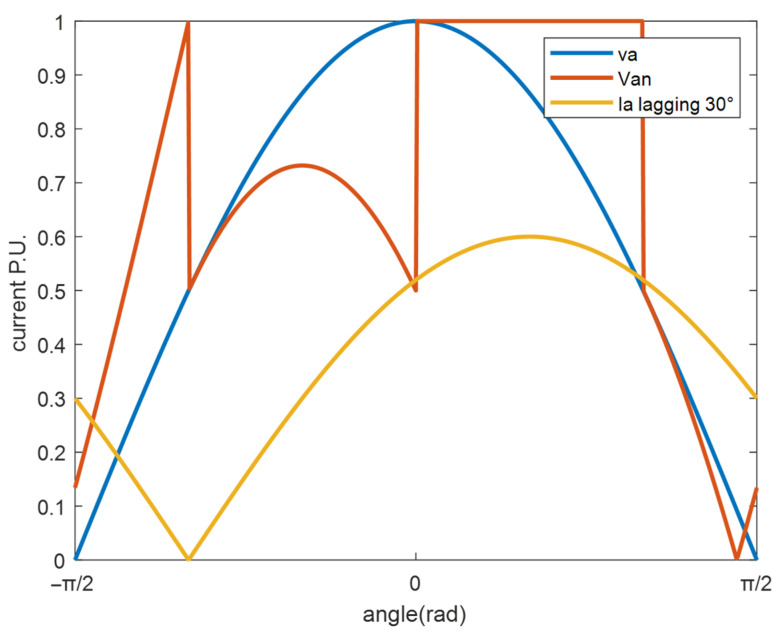
The modulated voltage waveform with DPWM2.

**Figure 8 micromachines-16-01128-f008:**
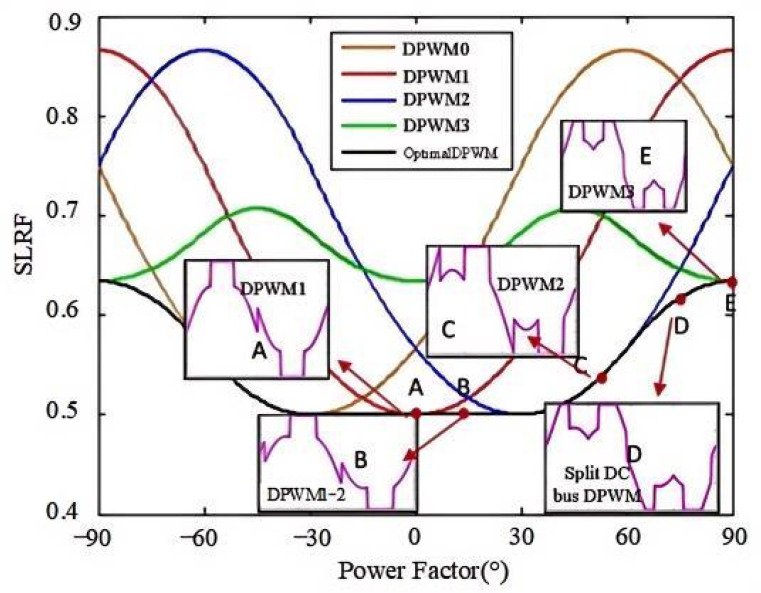
SLRF for different modulation modes under different power factors.

**Figure 9 micromachines-16-01128-f009:**
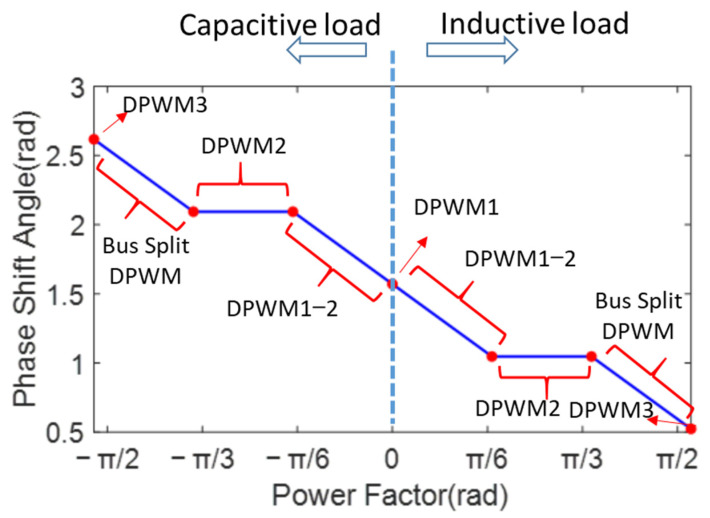
The relationship between the power factor angle and phase shift angle.

**Figure 10 micromachines-16-01128-f010:**
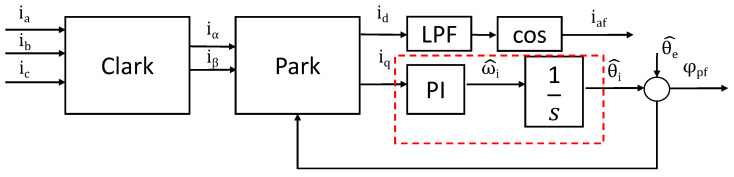
The control block for power factor angle extraction.

**Figure 11 micromachines-16-01128-f011:**
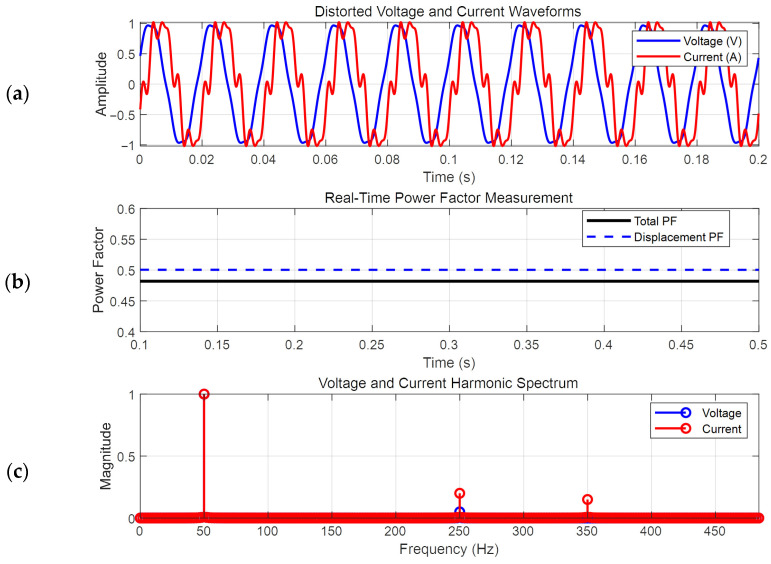
The control block for power factor generation. (**a**) Voltage and current; (**b**) total power factor and displacement power factor; (**c**) harmonics spectrum.

**Figure 12 micromachines-16-01128-f012:**
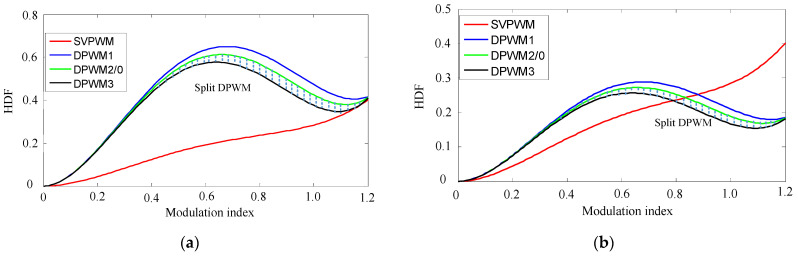
HDF functions. (**a**) All modulations at the same switching frequency. (**b**) The switching frequency is amplified by 1.5 times for all DPWM modes.

**Figure 13 micromachines-16-01128-f013:**
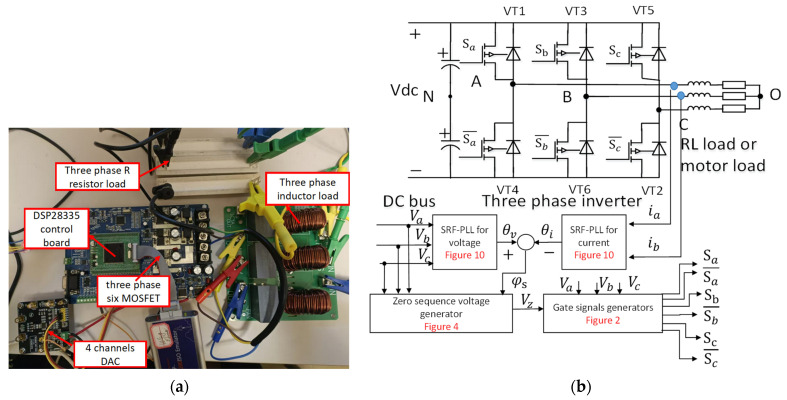
The experimental platform and systematic control block. (**a**) The three-phase inverter experimental platform. (**b**) The proposed modulation strategy control block.

**Figure 14 micromachines-16-01128-f014:**
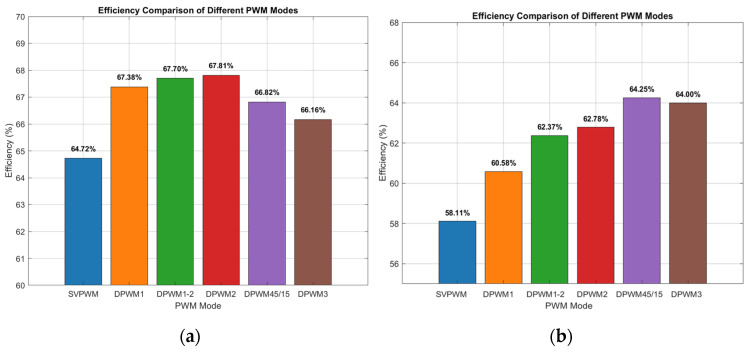
Inverter efficiency experiments under different modulation modes. (**a**) Current of 3A at 60 Hz; (**b**) current of 1.5 A at 400 Hz.

**Figure 15 micromachines-16-01128-f015:**
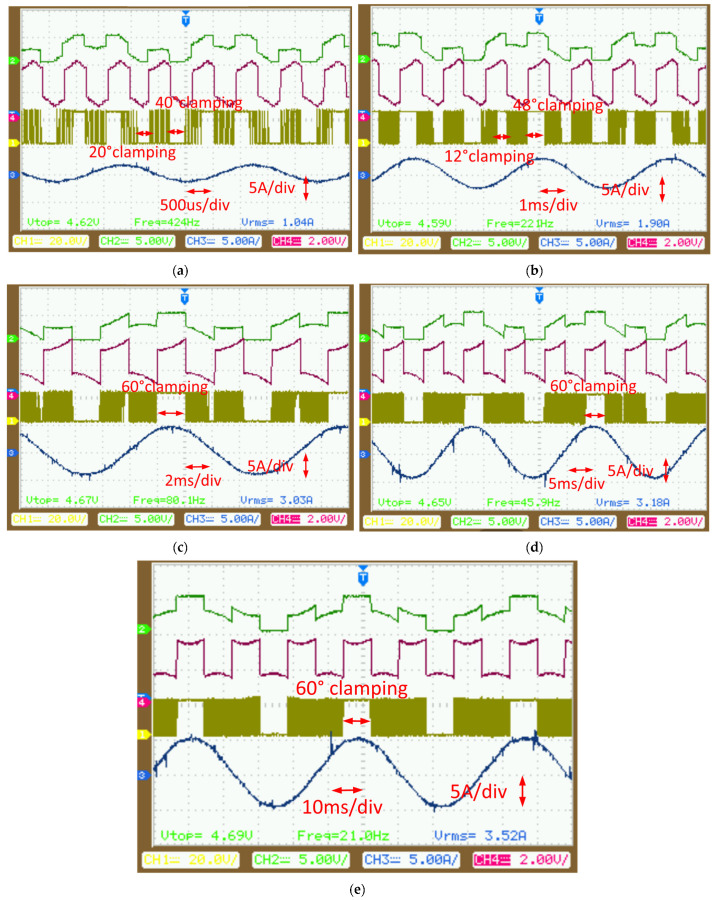
The optimal DPWM with different power factors: (**a**) power factor angle of 15°, (**b**) power factor angle of 30°, (**c**) power factor angle of 45°, (**d**) power factor angle of 72°, and (**e**) power factor angle of 80°.

**Figure 16 micromachines-16-01128-f016:**
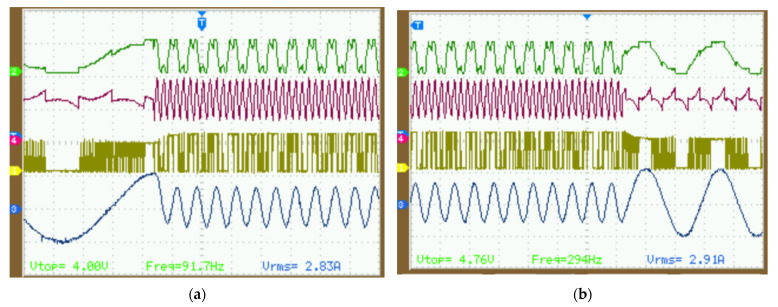
The step frequency dynamic test: (**a**) from 50 Hz to 300 Hz; (**b**) from 300 Hz to 80 Hz.

**Figure 17 micromachines-16-01128-f017:**
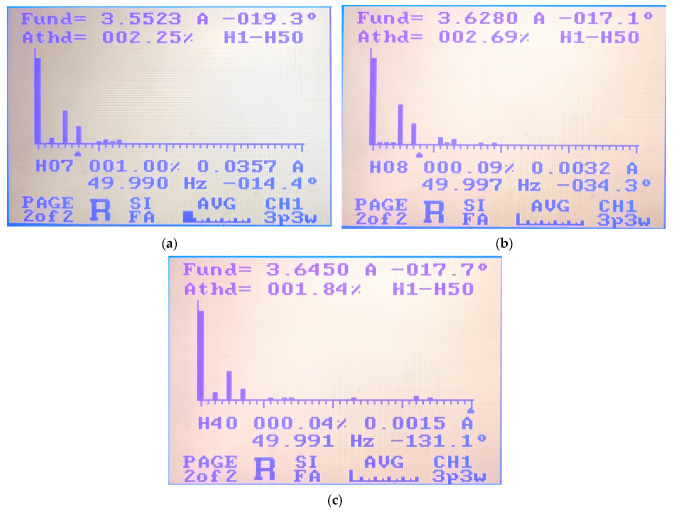
The THD under high modulation indexes: (**a**) 12 kHz—SVPWM; (**b**) 12 kHz—DPWM2; and (**c**) 20 kHz—DPWM2.

**Table 1 micromachines-16-01128-t001:** Different PWM modes’ loss analysis with 20 kHz switching at 3A-RMS 60Hz output current.

PWM Mode	Input Power (W)	Output Power (W)	Losses on Control Chip and Peripherals (W)	Losses on Micro Inverter (W)	Efficiency for Inverter
SVPWM	62.114	40.2	3.75	18.16	64.72%
DPWM1	63.5208	42.8	3.75	16.97	67.38%
DPWM1–2	60.2646	40.8	3.74	15.72	67.70%
DPWM2	59.5825	40.4	3.74	15.44	67.81%
DPWM45/15	59.26624	39.6	3.74	15.92	66.82%
DPWM3	62.42674	41.3	3.74	17.39	66.16%

**Table 2 micromachines-16-01128-t002:** Different PWM modes’ loss analysis with 20 kHz switching at 1.5A-RMS 400Hz output current.

PWM Mode	Input Power (W)	Output Power (W)	Losses on Control Chip and Peripherals (W)	Losses on Micro Inverter (W)	Efficiency for Inverter
SVPWM	16.35	9.5	3.76	3.09	58.11%
DPWM1	16.15	9.78	3.76	2.61	60.58%
DPWM1–2	16.19	10.1	3.76	2.33	62.37%
DPWM2	15.84	9.946	3.76	2.14	62.78%
DPWM45/15	15.75	10.12	3.76	1.87	64.25%
DPWM3	15.16	9.7	3.76	1.70	64.00%

## Data Availability

The original contributions presented in this study are included in the article. Further inquiries can be directed to the corresponding author.

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
