# Peer review of "Research on a Phase-Shift-Based Discontinuous PWM Method for 24V Onboard Thermally Limited Micro Voltage Source Inverters"

_micromachines, 2025, doi:10.3390/mi16101128_

Round 1
Reviewer 1 Report
Comments and Suggestions for Authors The manuscript shows very interesting topics. Although the DPWM methods are standardised, a smooth transition between these methods is not often used due to the necessity of measuring PF in case of distorted currents and voltages. I have only some not critical remarks and they are as follows:- In line 21 authors mention five intervals but they shows six intervals (30 to 60 is presented twice)
- In Figure 3 caption information about k = 0 for c. case is missed
- In Figure 4 arctangent or arctan function can be used not "artangent"
- In line 180 the it is mentioned thad DPWM1 and DPWM2 are described in Equation (5) but this equation shows the Clarke transformation and there is no information how DPWM1 or DPWM2 are constructed.
- The is no explanation what is the variable (greek fi) and what is a difference between fi given in (7)
- Below Figure 8 there is no stright information that black curve is equal to 0.5 because the interval where the phase is clamped to negative or positive dc voltage is following the change of the power factor thus the power losses are still the same.
- From lines 210 to 214 there is the information that the analysis is done for resisitve-inductive loads and the analysis for capacitive loads are obtained by symmetry. From that place the information that the paper focuses only RL loads is not true. Rewrite this part of the paper because in the following sections capacitive loads are also analysed.
- In lines 239 there is information that DPWM1 is depicted in Figure 7 however the figure caption shows that it is DPWM2, this is correct.
- It is not clear why the modulation index in Figure 12 is higher then 1.0 and equal to 1.2 . The maximum m = 1.15 whend SPWM is selected. Please rewise what should be the maximum m value.
Reviewer 2 Report
Comments and Suggestions for Authors
This paper proposes a discontinuous PWM method for inverter applications. The paper is well written and structured. The novelty of the paper is clearly presented, and the experimental validation is correctly described.
I have detected a single request to fulfill the quality needed for publication: please perform a mathematical analysis of the efficiency in terms of model. It means, considering the losses in the MOSFET and inductors, provide a mathematical development and prediction of the system efficiency. Then, such an analysis (and predictions) must be contrasted with the experimental efficiency measured in the platform. This procedure is needed for both design and viability analysis.
Reviewer 3 Report
Comments and Suggestions for Authors
The manuscript introduces a novel discontinuous PWM strategy with an adaptive phase-shift angle control for micro inverters. This is a relevant and timely topic, especially for applications that demand compact, efficient, and thermally constrained power converters. The combination of theory, simulation, and experimental validation makes the work technically strong and valuable.
That being said, there are several aspects that could be improved to make the paper clearer and more polished:
1. The English language needs revision. While the ideas are understandable, there are grammatical errors and awkward sentences that should be corrected to improve readability.
2. The terminology should be used consistently throughout the text (for example, “phase-shift angle,” “power factor angle,” “clamping mode”).
3. Some figures (notably Figures 5, 8, 15, and 17) are of low resolution and lack clear axis labels or units. Please improve their quality and enlarge the text for better readability.
4. Figure captions should be more descriptive, so the reader immediately understands what is being shown and compared.
5. In Section 2, the explanation of the proposed algorithm for zero-sequence voltage generation with phase-shift angle would benefit from a clearer, step-by-step description.
6. More details on the tuning parameters of the SRF-PLL are needed, along with an explanation of how it ensures stability under harmonic conditions.
7. The experimental results are convincing, but the discussion could better highlight how the proposed method compares with other state-of-the-art approaches.
8. In the harmonic analysis, please expand the discussion on how the proposed method balances efficiency and waveform quality.
Overall, the paper is technically solid and relevant for the journal’s audience. With improvements in language, figures, and presentation, the manuscript will become much clearer and stronger.
Round 2
Reviewer 1 Report
Comments and Suggestions for Authors
Thank you for taking my comments into account in the new version of the manuscript.
Reviewer 2 Report
Comments and Suggestions for Authors
The authors have correctly addressed my concerns.